# The Effect of Biological Treatment on Stress Parameters Determined in Saliva in Patients with Severe Psoriasis

**DOI:** 10.3390/medicina59040692

**Published:** 2023-03-31

**Authors:** Aleksandra Foks-Ciekalska, Jerzy Jarząb, Ewa Hadas, Elżbieta Świętochowska, Kamila Gumieniak, Wiktor Ciekalski, Andrzej Bożek

**Affiliations:** 1Clinical Department of Internal Diseases, Dermatology and Allergology in Zabrze, Medical University of Silesia, 40-055 Katowice, Poland; 2Department of Medical and Molecular Biology, Faculty of Medical Sciences in Zabrze, Medical University of Silesia, ul. Jordana 19, 41-808 Zabrze-Rokitnica, Poland; 3Department of Internal Medicine with Nephrology Subvision, Municipal Hospital No. 4, 44-100 Gliwice, Poland; 4Department of Internal Medicine, Municipal Hospital, 41-803 Zabrze, Poland

**Keywords:** psoriasis, stress, psychodermatology, saliva

## Abstract

*Background and objectives:* In psoriatic patients, stress is the most common aggravating factor. Despite the use of quality-of-life assessment questionnaires, diagnosing stress in psoriatic patients is not a flawless procedure. This study aimed to assess the usefulness of potential stress biomarkers in saliva for monitoring the treatment of psoriasis. *Materials and methods*: A total of 104 adult patients with severe psoriasis were included and randomly treated via biological treatment or symptomatic therapy: 84 received biological treatment, with 20 formed a control group receiving symptomatic therapy. The administered biological treatment was adalimumab, whilst in controls calcipotriol/betamethasone dipropionate topical gel and emollients were used. Patients were monitored monthly with a dermatological examination and the dispensing of a biological drug. During each of the four visits, the severity of the disease was assessed (PASI, BSA, and DLQI), and a sample of the patient’s saliva was taken. In all the participants, the saliva concentrations of immunoglobulin A (sIgA), α-amylase (sAA), and chromogranin A (CgA) were measured. *Results*: The majority of patients in both the study and control groups achieved clinical improvement, though favoring the group receiving biological treatment. The concentration of sIgA in the saliva was constantly increasing in the study group during subsequent visits (Fr = 27.26; *p* < 0.001). Meanwhile, there were no statistically significant changes in the control group during the same follow-up period (Fr = 6.66; *p* = 0.084). Levels of sAA underwent statistically significant changes in both groups (Fr = 58.02; *p* < 0.001—study group and Fr = 13.74; *p* = 0.003—control group). In the study group, a steady, statistically significant increase in sAA was observed from the first to the third visit. In the study group, a downward trend in CgA concentration was observed. In the control group, no significant differences in the level of CgA were obtained. Conclusions: sIgA, sAA, and CgA are potential markers of the severity of psoriasis and the associated stress reaction. Based on the presented observations, only sIgA and CgA seem to be valuable biomarkers for monitoring the effectiveness of the systemic treatment of psoriasis.

## 1. Introduction

According to psoriasis patients, stress is the main factor that triggers or exacerbates disease symptoms. As reported by other researchers, stress was responsible for the recurrence or aggravation of psoriasis in about 70.0% of patients [1,2].

The relationship between stress and psoriasis can be described as a vicious circle effect: on the one hand, the psychophysiological state of the patient is involved in the development and course of the disease, on the other hand, the disease generating severe stress and anxiety can be the cause of the development of emotional disorders and mental illness [3]. Psoriasis significantly affects patients’ quality of life (QOL), due to several issues, including its chronic, disabling, and disfiguring nature, with symptoms, such as discomfort and associated stigma [4]. The Dermatology Life Quality Index (DLQI) was created in the 1990s and has since become a widely used tool for assessing the impact of a dermatological disease on a patient’s life [5,6]. Conventionally, the severity of disease has been assessed based on clinical symptoms: body surface area (BSA) and lesion severity (PASI—Psoriasis Area and Severity Index).

Unfortunately, these and other similarly accurate instruments do not capture the actual severity of the disease from the patient’s point of view and the impact on their lives that may be much greater than the visible severity measured by these tools [7]. The DLQI questionnaire includes 10 questions related to the patient’s daily life, and it assesses the impact of the disease and related treatment on such spheres as the patient’s wellbeing, work and school, family relationships, or sex life. The sum of the answers gives a score from 0 to 30, where a higher score indicates a greater impact of psoriasis on the patient’s daily life [8].

Despite the QOL assessment questionnaires, diagnosing stress and anxiety in psoriatic patients is a problematic process. There are no laboratory tests to exclude doctors’ doubts about the real impact of the disease on the patient’s mental state and the subsequent effect of treatment on improving this area [9]. Therefore, this study will analyze saliva components in the search for relevant biomarkers in the diagnosis of stress. The parameters we focused on were: sAA, sIgA, and CgA.

The study aimed to assess the usefulness of the biomarkers mentioned above in saliva for monitoring the treatment of psoriasis. The originality of this observation was that previous studies presented only the changed value of the concentrations of these parameters at the beginning without assessing their possible variability during therapy and obtaining a clinical effect.

## 2. Materials and Methods

### 2.1. Study Design

Our study was an observational, one-centre study with patient randomisation to the study or control groups, which was performed from March 2020 to December 2021 in the Dermatological Department. The Bioethical Committee of the Medical University of Silesia in Katowice approved the present study (consent no. PCN/0022/KB1/13/20).

### 2.2. Patients

The diagnosis of psoriasis was based on a dermatological assessment and PASI scales. Ultimately, 104 patients with psoriasis were included in the study based on the inclusion criteria. The final inclusion criteria were: age > 18 years; suffering from severe disease, as determined by an assessment of the severity of the psoriatic process based on a PASI score greater than 18, and either a DLQI score greater than 10 or a BSA score greater than 10, with signed consent.

Exclusion criteria were female patients who are pregnant or breast-feeding or considering becoming pregnant during the study, evidence of dysplasia or a history of malignancy (other than a successfully treated basal cell carcinoma), a history of listeriosis, histoplasmosis, a chronic or active hepatitis B infection, human immunodeficiency virus (HIV) infection, immunodeficiency syndrome, chronic recurring infections or active tuberculosis (TB), a history of moderate to severe congestive heart failure, a history of central nervous system (CNS) demyelinating disease or neurologic symptoms suggestive of CNS demyelinating disease, diagnosis of lupus erythematosus, and pancytopenia and aplastic anemia.

Patients were randomly selected to therapy with biological treatment or only to symptomatic pharmacoterapy in a 4:1 ratio. In this way, 84 adult patients received biological treatment, and 20 adult control patients received only symptomatic therapy. All patients signed consent.

The study group consisted of 29 women and 53 men. The control group included eight women and twelve men. The result of the chi-square test (χ^2^ = 0.016; *p* = 0.899) indicates that the gender structure in the two groups is not significantly different.

The ages of those in the study group ranged from 29–71, with an average of 52.4 ± 11.7 years, and those in the control group ranged from 25–75, with an average of 50.7 ± 12.2 years. The age distribution in both groups was not significantly different from the normal distribution, as verified by the Shapiro-Wilk test, yielding *p* = 0.746 in the study group and *p* = 0.215 in the control group. Therefore, a comparison of age in the two groups was made using the two independent sample t-tests, obtaining t = 0.545 and *p* = 0.587. The result indicates that there is no statistically significant difference.

### 2.3. Treatment

The administered biological treatment was adalimumab. The starting dose was 80 mg, subcutaneously. Then, one week after the initial dose, a dose of 40 mg every two weeks was used.

In controls, combination of calcipotriol/betamethasone dipropionate topical gel and emollients were used. When necessary, patients in the study group also used this topical treatment. Synthetic corticosteroid/synthetic vitamin D3 combination is a safe and effective way to treat plaque psoriasis of mild to moderate severity as a first-line topical treatment [10,11].

### 2.4. Procedures

#### 2.4.1. Observation

Patients were monitored monthly at the centre with a dermatological examination and the dispensing of a biological drug. At each of the four visits, the severity of the disease was assessed (PASI, BSA, DLQI), and a sample of the patient’s saliva was taken. In the study group, the first visit took place at the time of the inception of treatment, and the next three were performed during biological therapy. Each time, 3 mL of saliva was taken with Salivettes (Sarstedt^®^, Nümbrecht, Germany). Saliva was taken in the morning. Patients were instructed to remain fasting and advised to avoid physical activity on the day of collection. Collected material was transported under appropriate conditions to the Department of Molecular Biology at the Medical University of Silesia in Katowice for analysis. The material was stored in the refrigerator at 4 °C.

#### 2.4.2. Assessment of Disease Severity

The Psoriasis Area and Severity Index (PASI) takes into account both lesion severity and the affected area in the form of a single score from 0 to 72.

Body Surface Area (BSA) index determines the percentage of body surface area occupied by psoriatic lesions, ranging from 0 to 100. The calculation of the BSA value uses the rule of nines, which was originally used in estimating the area of burns.

The Dermatology Life Quality Index (DLQI) includes 10 questions related to the impact of the disease on various aspects of daily life over the previous week.

#### 2.4.3. Laboratory Tests

##### Measurement of α-Amylase Activity

A static method [12] using a kit (Aqua-Med., Łódź, Poland) was used for the measurement of α-amylase activity; 2-chloro-4-nitrophenylmaletrioside is a substrate in this method, and the reaction was performed in the MES buffer at pH of 6.9 and 37 °C. The spectrophotometric reading of the resulting coloured product was performed at 405 nm. Saliva samples were diluted 100 times with 0.9% NaCl. The obtained results were expressed in the units of salivary α-amylase activity (U/mL). The margin of error of the method was 4.1%.

##### Estimation of Chromogranin A Levels

Concentrations of chromogranin A were estimated using a commercial kit from Cisbio Bioassays—Codolet, France, cat. no.017, according to the manufacturer’s instructions. A calibration curve was prepared to determine the concentrations of the test samples using the standards included in the kit. The spectrophotometric reading was carried out using the Universal Microplate Spectrophotometer- µQUANT from BIO-TEK INC (Bio-Tek World Headquarters, City of Industry, CA, USA), at 450 nm. The processing of results was carried out using the KC Junior computer program (Bio-Tek, City of Industry, CA, USA). The sensitivity of the kit was 7 ng/mL, and the margin of error of the method was 12%.

##### Measurement of Salivary Secretory Immunoglobulin A

The sIgA ELISA Kit cat. no. K8870 from Immunodiagnostic AG (Bensheim, Germany) was used for the measurement of sIgA concentration. The analytical procedure followed the manufacturer’s instructions included in the kit. The spectrophotometric reading was carried out using a μQuant reader (Bio-Tek, City of Industry, CA, USA), while the results were processed using KC Junior software (Bio-Tek, City of Industry, CA, USA). The method’s sensitivity was 2.5 μg/mL, the margin of error of the method was 5.3%, and the reproducibility was 5.6%.

### 2.5. Statistical Methods

In this study, six variables were analyzed. The first three (PASI, DLQI, BSA) are expressed on an ordinal scale, and the remaining three (α-Amylase, Chromogranin A, Secretory Immunoglobulin A) are expressed on an interval scale.

To assess the significance of differences in all analyzed variables over the course of treatment (four visits), the non-parametric Friedman test was used.

The use of this test for variables expressed on the interval scale followed the negative results obtained with the sphericity test, as well as the results of assessing the normality of the distributions with the Shapiro-Wilk test.

In the case of obtaining statistical significance in the Friedman test, the Conover post-hoc test was applied.

A non-parametric Mann-Whitney test was used as part of the statistical analysis to compare the results obtained at subsequent visits between the study group and the control group.

Comparison of the two groups by gender was performed using the chi-square test with Yates correction.

Comparison by age of subjects in both groups was made using the two-tailed two independent sample *t*-test due to the hypotheses of normality of age distribution in both groups.

## 3. Results

### 3.1. Efficacy of Therapy

During observation, the majority of patients in the study and control groups achieved clinical improvement, favoring the group receiving biological treatment. Changes in DLQI score during the conducted observation are presented in Figure 1.

### 3.2. Variation of Biomarkers during Treatment

The concentration of immunoglobulin A in the saliva was found to be constantly increasing in the study group during subsequent visits and biological treatment (Friedman’s test: Fr = 27.26; *p* < 0.001). There were no statistically significant changes in the control group during the same follow-up period (Fr = 6.66; *p* = 0.084) (Figure 2). Levels of alpha-amylase (Figure 3) throughout the observation period underwent statistically significant changes in both groups (Fr = 58.02; *p* < 0.001—study group and Fr = 13.74; *p* = 0.003—control group). In the study group, a steady, statistically significant increase in alpha-amylase was observed from the first to the third visit. The difference between the third and fourth visit was not statistically significant. In the control group, a steady increase in alpha-amylase was observed only from the second visit to the end of the observation period. During the first visit, the level of alpha-amylase was significantly higher in the control group, and in subsequent stages, the differences between the groups were not statistically significant. In the study group, a downward trend in Chromogranin A concentration was observed, with the exception of the second visit, where its level reached the highest value. Meanwhile, in the control group, no significant differences in the level of Chromogranin A were obtained (Figure 4).

## 4. Discussion

The expected observations of the conducted study are the clinical improvement of patients in the biological treatment group and, to a lesser extent, in the control group. All the parameters we determined underwent significant changes during therapy. Differences in parameter concentrations were associated with improvements in patients’ clinical condition (PASI, BSA) and with a reduction in the impact of the disease on their quality of life (DLQI). This is in line with many observations that prove the crucial role of biological therapy in psoriasis treatment [13].

However, the study’s main aim was to evaluate the selected biomarkers and their possible role in monitoring the effects of treatment. Immunoglobulin A, alpha-amylase and chromogranin A were selected as markers of stress and its impact on the course of psoriasis based on previous observations [14]. The present study confirmed the effect of treating severe psoriasis on stress parameters determined in saliva.

The role of stress in the severity of psoriasis has been confirmed in many studies, as in other somatic diseases [15,16,17,18]. Researchers in other fields have also tried to determine these parameters in disease entities, such as ulcerative bowel syndrome, colitis ulcerosa, neuropsychiatric diseases, and oral diseases [19,20,21]. Unfortunately, most of them did not determine the parameters before the inclusion and during the course of treatment, which makes it difficult to compare the results with our study.

Stress induces the physiological activation of specific areas of the nervous system, both central and peripheral. Activation of these areas results in further stimulation of the hypothalamus and brainstem, activation of the hypothalamic-pituitary-adrenal axis, and of the autonomic nervous system. These systems are in close communication with the immune system, which means that they play a key role in the pathogenesis of stress-related disease entities [22]. During the aforementioned compensatory reactions of the organism, numerous organic chemical compounds are produced and secreted into body fluids [23]. Saliva includes markers and a material that allows simple sampling and storage [24], therefore, it seems that its role in diagnostic laboratory investigations is still underestimated. There are few reports on biomarkers associated with psoriasis in saliva [25].

Finding in vitro parameters that adequately correlate with stress and its reduction during psoriasis treatment is a much more difficult task. Currently, we can see only single publications confirming their importance [26,27]. These include the biomarkers presented also in this study. Unfortunately, most of these papers focused on static values without analyzing their variability under the influence of treatment. The presented research does this for the first time. Noticeable trends of changes in the observed biomarkers, although extant, did not always correlate with the author’s expectations.

The state of the immune system is closely related to the stress response. Under the influence of a stress stimulus, immunoglobulin production is inhibited [28]. IgA is a class of antibodies found in the mucous membranes, which is the body’s first defence factor in contact with an infectious pathogen or an allergen [28]. The salivary IgA levels increased with increasing age up to 60 years and then decreased [29]. sIgA peaks in the morning and then gradually declines till the evening [30]. Psychological factors have a direct effect on the concentration of immunoglobulin A in saliva. As a result of positive stimuli, its concentration increases, while a depressed mood and stress cause a reduction in its amount. [28,31]. Researchers have proven that, in animals, both chronic and acute stress reduced salivary IgA levels [32]. It has been proven that chronic stress in humans is related to the activation of the HPA axis and to decreased immune system activity (measured by a decrease in salivary IgA and lysozyme concentration) [23,28]. A significant relationship between perceived stress, depressive symptoms, and anxiety with a low level of salivary immunoglobulin A was also confirmed [33]. However, some studies [12] showed a significantly decreased level of psoriatic compared to controls.

In our study, sIgA levels significantly increased during biological treatment, which may constitute proof that the psoriasis itself determines the maintenance of low levels of this immunoglobulin. However, relevant treatment and stress reduction significantly increase its levels.

Salivary alpha-amylase is the main digestive enzyme of the oral cavity, which, apart from the hydrolysis of starch and glycogen, also has an immunological function, protecting the oral cavity from microorganisms [34]. sAA is a recognized marker of sensitivity to adrenergic stimulants [35]. In healthy individuals, alpha-amylase levels are lowest in the early morning and highest in the late afternoon [36]. The level of sAA shows no changes with age [37]. The concentration of sAA increases during a stress response, especially an acute example, making it a valuable marker of this condition [38,39]. The changes of sAA as a salivary biomarker of acute stress and anxiety during dental therapy was confirmed by Jafari et al. [40].

On the other hand, Skutnik-Radziszewska et al. [41] confirmed that the activity of salivary amylase in psoriasis patients was visibly lower than in a control group. Their research proved that, with the progression of the disease, there is a loss of secretory function of the parotid and submandibular salivary glands. This condition results in a reduction in salivary secretion and also reduced salivary amylase activity and overall protein concentration [41]. This may explain why amylase concentrations increased as the patient was treated and improved.

In other studies, the concentration of alpha-amylase was either significantly higher [42] or not significantly higher in patients with psoriasis compared to the control group [43,44].

Chromogranin A is known a major soluble protein in adrenal medullary chromaffin granules and adrenergic neurons, and it coreleases with catecholamines, which are regarded to be a good index of sympathetic activity [45,46]. Specifically, salivary CgA is considered as sensitive index of psychological stress, while it does not respond to physical stress [47]. An important advantage of salivary CgA determination, compared to the aforementioned salivary stress biomarkers, is that CgA levels are not affected by time of day [40]. Baseline chromogranin A concentration decreases with age [48]. Salivary CgA concentration responds quickly to changes in psychological tension [47]. In our study, a decreasing trend in chromogranin A levels was observed in the study group, in contrast to the control group, where no significant differences were observed. This could be a valuable monitoring biomarker for psoriasis treatment and stress reduction. However, it requires further research.

Nijakowski et al. [49] showed that the TNF blockade per se does not determine changes in the parameters measured in saliva. In their study, only patients who achieved clinical improvement during adalimumab treatment of inflammatory bowel diseases showed changes in the determined parameters. The results of the others remained unchanged despite administration of the drug [D]. Further studies are needed to evaluate the effects of IL-17 and IL-23 inhibitors on these parameters.

The main limitations of this research are the focus only on severe psoriasis and the relatively small control group, which also could influence effect of randomization. The clinical severity decides factors concerning inclusion, and all analysed parameters were of secondary importance. However, the authors thought that severe psoriasis and intense therapy could optimally influence changes in biomarkers. To show the variability of biomarkers depending on the scale of therapeutic intervention, the control group consisted of patients with psoriasis (and not healthy ones) with topical treatment. Based on this assumption, we proved their more significant variability in the study group. This means that the analyzed biomarkers may be helpful in assessing the effectiveness of psoriasis treatment combined with stress reduction. However, this requires further research.

Studies with wider salivary profiling are needed to confirm the preliminary reports [27]. Moreover, investigations focused on observing differing concentrations’ changes of these saliva indicators during treatment are also necessary. More studies on potential salivary prognostic biomarkers in patients with psoriasis are still needed as they might be helpful detectors of psoriasis severity, disease progression and treatment effects. The identification of relevant biomarkers may provide valuable diagnostics in the future.

## 5. Conclusions

The analysed salivary immunoglobulin A, chromogranin A, and alpha-amylase are potential markers of the severity of psoriasis and the associated stress reaction. Based on the presented observations, only sIgA and CgA seem to be valuable biomarkers for monitoring the effectiveness of the treatment of psoriasis. However, this requires further research.

## Figures and Tables

**Figure 1 medicina-59-00692-f001:**
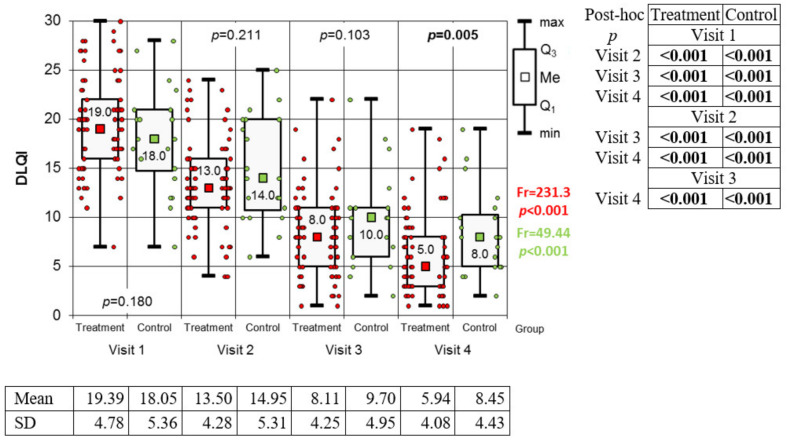
Changes in DLQI values from one to four visits in both groups (results of Friedman test and results of Conover post-hoc test). Legend: DLQI: Significant differences in DLQI values throughout the observation period in both groups (Friedman test: Fr = 213.3; *p* < 0.001—study group and Fr = 49.44; *p* < 0.001—control group). The results of the Conover post hoc test used in this situation showed that the DLQI values in both groups were subject to a continuous, statistically significant decrease from visit to visit.

**Figure 2 medicina-59-00692-f002:**
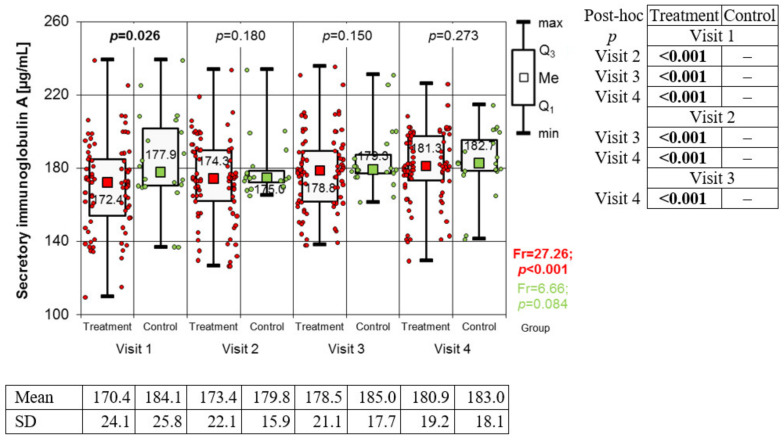
Changes in immunoglobulin A value from one to four visits in both groups (results of Friedman test and results of Conover post hoc test).

**Figure 3 medicina-59-00692-f003:**
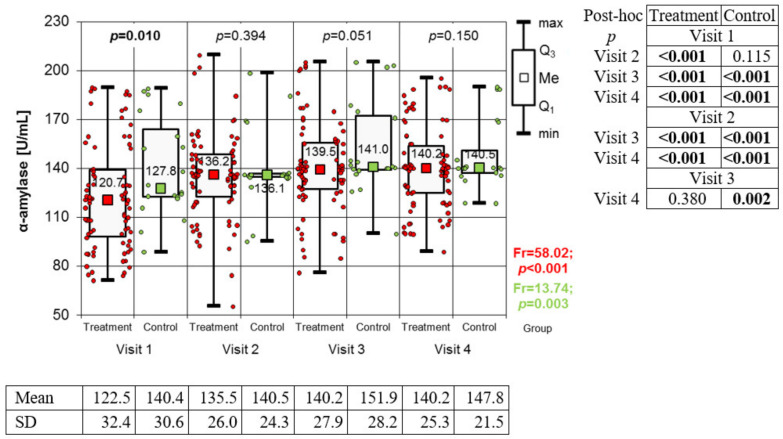
Changes in α-amylase values from visit one to four in both groups (results of Friedman test and results of Conover post hoc test).

**Figure 4 medicina-59-00692-f004:**
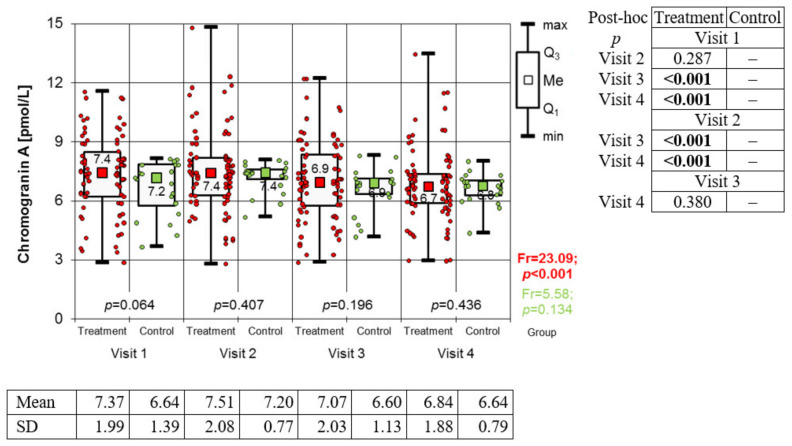
Changes in Chromogranin A value from one to four visits in both groups (results of Friedman test and results of Conover post hoc test).

## Data Availability

The data presented in this study are available on request from the corresponding author. The data are not publicly available due to ethical restrictions.

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
