# Peer review of "The Effect of Biological Treatment on Stress Parameters Determined in Saliva in Patients with Severe Psoriasis"

_medicina, 2023, doi:10.3390/medicina59040692_

Round 1
Reviewer 1 Report
An interesting study, that should be revised:
Lines 51-54, you need some references, such as: doi: 10.1111/dth.13185. and doi: 10.3390/pharmaceutics14020294.
The critical points of the study can be identified in the sample's smallness and in the observation time's short duration. The study is a single-center observational study; therefore, the samples examined come from a single care setting; this aspect reduces the clinical applicability of the study results. Historically, psoriasis was viewed as a complex immune-mediated disease in which T lymphocytes, dendritic cells, and cytokines (interleukin [IL] 23, IL-17, and tumor necrosis factor [TNF]) play a central role; however, there are few reports on biomarkers associated with psoriasis in saliva. The markers examined are too nonspecific, as they are altered in other stress-related pathologies. Exclusion criteria exclude obesity, psychiatric illness, and all other autoimmune conditions except lupus. The endpoint considered would be more interesting, with a larger sample, longer observation times, and more stringent inclusion and exclusion criteria.
Author Response
Dear Reviewer,
We appreciate reviewers’ thoughtful comments which have substantially improved the content and clarity of this piece. Please find below more detailed responses to the comments.
Please see the attachement.
Best regards,
Authors

Reviewer 2 Report
Foks-Ciekalska et al. present data on stress-related salivary components of psoriasis patients before and during biologic and local treatment. Salivary IgA, salivary amylase and chromogranin are presented as potential biomarkers for psoriasis-related stress.
Following comments:
The control group of 20 patients with local betamethasoine/calcipotriol treatment is rather small.
All parameters (PASI, DLQI and salivary parameters are divergent between the two study groups before starting treatment though groups were randomly selected.
The results of decreasing PASI and DLQI during treatment are really not surprising and may be shortened. In discussion, clinical improvement is regarded as „primary observation“.
Have all patients starting the study been followed up?
Ranges are very broad, especially for BSA and sIgA. This should be commented.
How specific are these results for psoriasis? Any disease controls?
Does TNF blockage per se influence any of the tree parameters irrespective of PsO activity. This should be discussed.
How would other cytokine blockers (IL-17, IL-23) affect these parameters?
Any comedication and potential influence on salivary parameters?
Do these parameters show any daily changes? Any influence of age of patients?
Author Response

(The authors gave the same response as above.)
